

# Technical note: A comparison of methods for estimating coccolith mass

Celina Rebeca Valença[1], Luc Beaufort[2], Gustaaf Marinus Hallegraeff[3], Marius Nils Müller[4]

[1]Department of Oceanography, Federal University of Pernambuco, Recife, 50740-550, Brazil
5   [2]Aix-Marseille University, CNRS, IRD, INRAE, CEREGE, Aix-en-Provence, France
[3]Institute for Marine and Antarctic Studies, University of Tasmania, Private Bag 129, TAS 7001, Australia
[4]Macau Environmental Research Institute, Macau University of Science and Technology, Macau

*Correspondence to*: Marius Nils Müller (mmnils@must.edu.mo)

**Abstract.** The fossil record of coccolithophores dates back approximately 225 million years and the production of their
10   calcite platelets (coccoliths) contributes to the global carbon cycle over short and geological time scales. Variations in
coccolithophore parameters (e.g., community composition, morphology, size and coccolith mass) have been used as
paleoproxy to understand past oceanographic conditions. Coccolith mass has been frequently estimated with different
methods with electron microscopy the most applied. Here, we compared the electron microscopy (EM) method with the
Coulter Multisizer (CM) (i.e., electric field disturbance) and Bidirectional Circular Polarization (BCP) methods to estimate
15   coccolith masses in controlled laboratory experiments with two ecotypes of *Emiliania huxleyi*. Average coccolith mass
estimates were in good agreement with literature data. However, mass estimates from CM were slightly overestimated
compared to EM and BCP estimates and a correction factor ($c_f = 0.8$) is suggested to compensate for this discrepancy. The
relative change in coccolith mass triggered by morphotype specific structures and environmental parameters (i.e., seawater
carbonate chemistry) was suitably captured by each of the three techniques.



# 1 Introduction

The ocean constitutes a crucial part in the biogeochemical cycling of the Earth's elements and represents an important interface between the atmosphere and lithosphere. Phytoplankton are integral to the biogeochemical cycling of the ocean, as they convert inorganic carbon and nutrients into organic compounds through photosynthesis, serving as the foundation for the marine food web and facilitating the transfer of carbon from the atmosphere to the ocean's interior (Litchman et al., 2015). Coccolithophores have been present in Earth's oceans for approximately 225 million years (Not et al., 2012). These organisms produce calcite platelets called coccoliths and sequester significant amounts of calcium carbonate ($CaCO_3$) into sea-floor sediments, contributing to the global cycle of carbon and other elements (Sviben et al., 2016). The geochemical composition of coccoliths can provide valuable insights into the biogeochemical cycles of divalent cations such as magnesium (Mg), strontium (Sr), and barium (Ba) (Müller et al., 2014; Bolton et al., 2016). The assemblages of coccoliths found in sediments accurately represent the living coccolithophore community in the upper mixed ocean (Jin et al., 2019). Changes in morphology, mass and geometry of the intracellularly produced coccoliths has been used to understand palaeoceanographic conditions and investigate the potential influence of this phytoplankton group on climate dynamics and $CO_2$ fluxes (Renaud and Klaas, 2001; Fielding et al., 2009; Beaufort et al. 2011). Changes in environmental parameters (e.g., salinity, nutrient availability, temperature, carbonate chemistry) can affect the physiology and cell size of coccolithophore and, consequently, has the potential to alter the mass of coccoliths (Fielding et al., 2009; Jin et al., 2016; Horigome et al., 2014; D'Amario et al., 2018). It has been proposed that coccolith size changes occur proportionally to coccosphere or cell size (Aloisi, 2015; Müller et al., 2021) but this "coccolithophore size rules"- hypothesis has been recently challenged and further refined (Suchéras-Marx et al. 2022).

Coccoliths have been categorized into holo- and heterococcoliths, depending on life cycle phases and the associated extra- and intracellular biomineralization processes, respectively (Rowson et al., 1986; Young and Henriksen, 2003). Heterococcolith formation occurs in a special cellular compartment within the coccolithophore cell and completely formed coccoliths are extruded to the cell surface and arranged to compose the coccosphere. The underlying physiological and biogeochemical mechanisms of coccolith formation have been partly revealed (e.g., Mackinder et al., 2011; Mejía et al., 2018) and several hypotheses for the ecological/cellular function of coccolithophore calcification and genesis have been presented (Monteiro et al., 2016; Müller, 2019).

The coccolithophore species *Emiliania huxleyi*, more recently based on genetic arguments also referred to as *Gephyrocapsa huxleyi* (Bendif et al., 2023; Wheeler et al., 2023), is amongst the numerically most abundant and geographically distributed representatives of this functional group in the modern ocean, being an important model species for physiological and biomineralization studies (e.g., Henderiks and Pagani, 2008; Triantaphyllou et al., 2010; Poulton et al., 2011; Müller et al., 2012; Hoffmann et al., 2015; Faucher et al., 2020). In the Southern Ocean, the most dominant ecotypes of *E. huxleyi* are A and BC with distinct differences in their cellular physiology and morphological coccolith structure. The distribution and





physiological performance of these ecotypes is determined by temperature, the Antarctic Polar Front and seawater carbonate
chemistry (Mohan et al., 2008; Winter et al., 2014; Cubillos et al., 2007; Müller et al., 2015).

The microscopic nature of coccoliths impedes a direct measurement of its mass and several methods have been applied to indirectly estimate single coccolith masses. Electron microscopy is a powerful technique that uses a beam of electrons to visualize and analyse particles at high resolution. The utilization of electron microscopy has proven to be a valuable technique for measuring the geometry (length, width and thickness) and estimating the mass of coccoliths with the
application of geometric constants for specific coccolithophore species and coccolith structures (Young and Ziveri, 2000). The detailed imaging capabilities of electron microscopy reveal the intricate patterns, features and morphology of coccoliths from sediment, ocean and laboratory samples (Saruwatari et al., 2011). However, electron microscopy requires specialized equipment and experience to differentiate between different coccolith species and to identify any variations in size and morphology, while at the same time sample preparation can be time-consuming and complex.

Another geometric approach to estimate the coccolith mass, especially in laboratory experiments, is the analysis of coccoliths using the Coulter Multisizer principle (Müller et al., 2012; 2021). This method is a commonly employed technique in particle measurement, utilizing electrical impedance to ascertain the volume distribution of particles present in a conductive liquid. The Coulter Multisizer device is comprised of a small orifice with an applied electric field through which the particle suspension (sample) is passed. As the particles flow through the orifice, they disrupt an electric current that is
applied across the orifice. The number, amplitude and duration of disruptions (or pulses) are directly related to the number and volume the analysed particles. The Coulter Multisizer principle provides a rapid method to count coccolithophore cells, coccoliths and estimate extracellular $CaCO_3$ content from laboratory experiments where sufficient sample material is available (Müller et al., 2021; Fan et al., 2022).

The optical properties of coccoliths (i.e., calcite) allow the application of polarizing or polarized light microscopy for
analysis. Utilizing polarized light is an advanced technique that enhances the contrast of images obtained from birefringent materials, demonstrating a heightened level of sensitivity and can be effectively adjusted for both quantitative and qualitative studies. Applications of polarized light microscopy facilitates the recognition and characterization of diverse coccolithophore species (Gordon and Du, 2001; Bollmann, 2014). This method allows the observation and documentation of the shape, size, and arrangement of coccoliths. As a result, it yields valuable taxonomic data and coccolith mass estimations when
appropriately calibrated (Beaufort, 2005; Beaufort et al., 2014; Meier et al., 2014).

All three described methods have been applied in coccolithophore studies but no direct comparison has been reported. Here, coccolith mass estimations derived from electron microscopy, Coulter Multisizer and cross-polarized light methodologies are compared using samples from controlled laboratory experiments with two Southern Ocean coccolithophore ecotypes of the species *Emiliania huxleyi*.




## 2 Material and Methods

Four Southern Ocean strains of *E. huxleyi* (strains SO 5.14 and SO 5.30 of ecotype A and SO 5.11 and SO 8.15 of ecotype BC, isolated in 2007 by S. Cook) were grown under nutrient replete batch-culture conditions at 14°C in 0.2 µm filtrated natural seawater with a salinity of 35 and a continuous photon flux density of 100 to 115 µmol photons $m^{-2}$ $s^{-1}$, assuring

desynchronization of the cellular division cycle and independence of sampling time and cellular volume (Müller et al., 2008; 2021). Detailed culture conditions of the experiments (conducted in 2012/2013) are described in Müller et al. (2015). In summary, seawater carbonate chemistry was manipulated for each strain in triplicate treatments to generate a $pCO_2$ gradient ranging from 296 - 1683 µatm, corresponding to a $pH_{(total\ scale)}$ gradient from 8.17 to 7.48. Samples for EM and BCP microscopy were pooled for each triplicate treatment and filtered onto polycarbonate filters (0.8 µm pore size) and then dried

at 60°C pending analyses. Samples for coccolith volume were processed directly using a Coulter Multisizer™ 4 (Beckman Coulter Life Sciences) equipped with a 30 µm aperture, calibrated with 5 µm Latex particles (NIST traceable standard) and following established protocols (Fan et al., 2022; Faucher et al., 2017; Müller et al. 2012; 2017; 2021).

### 2.1. Scanning Electron Microscopy

Sample filters were sputter coated (Gold-Palladium) and observed on a Hitachi SU-70 field emission scanning electron microscope at the Central Science Laboratory of the University of Tasmania. Images were taken at 1500x magnification and analysed using the 'Image J' software (Schneider et al., 2021). Image J was calibrated using the size bars of the images. Only single coccoliths lying "face up" were analysed for Distal Shield Length (DSL), Distal Shield Width (DSW), Central Area Length (CAL), Central Area Width (CAW), Distal Shield Area (DSA) and the Central Area Area (CAA) (Figure 1). DSA

and the CAA were calculated assuming that both areas are resembled in a standard elliptical form. An average number of 100 and 80 coccoliths were analysed for each geometric parameter of ecotype A and BC, respectively (see supplementary data).



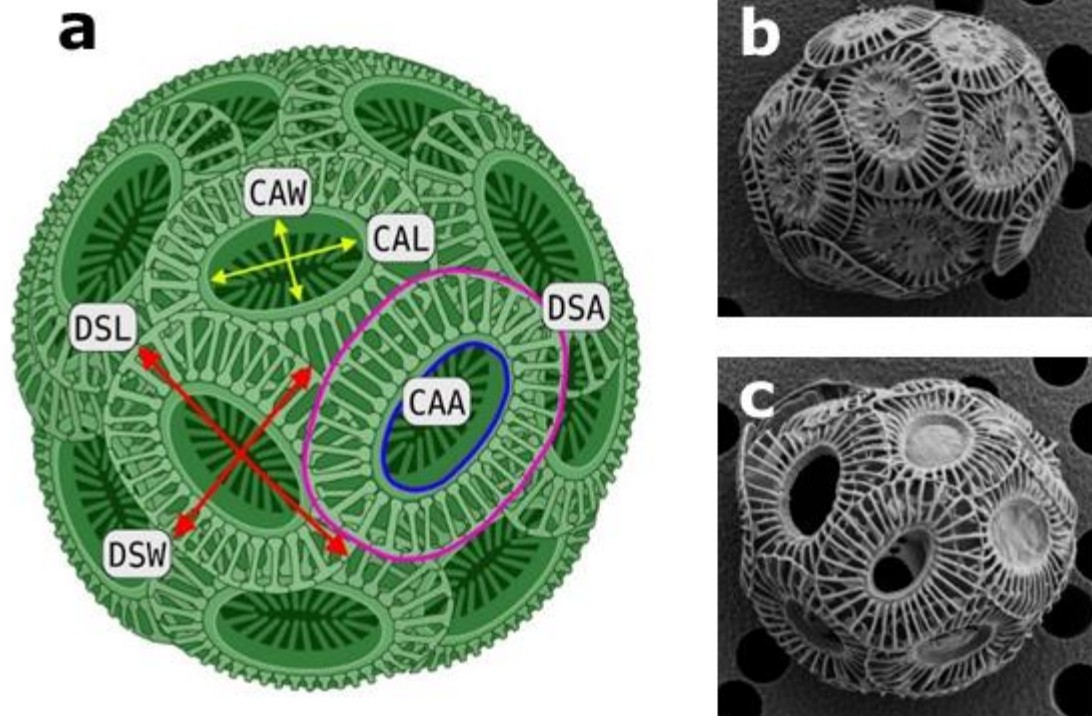

**Figure 1: Coccospheres of *Emiliania huxleyi*.** **(a) Schematic illustration of a coccosphere, showcasing the evaluated geometric measurements using scanning electron microscopy (DSL = Distal Shield Length, DSW = Distal Shield Width, DSA = Distal Shield Area, CAL = Central Area Length, CAW = Central Area Width, CAA = Central Area Area). Representative EM images of Southern Ocean *E. huxleyi* ecotype A (b) and BC (c).**

## 2.2. Polarizing Microscopy

Sample filters were affixed using a UV optical mounting medium (Norton Optical 74). Employing an automated optical microscope (Leica DM6000) equipped with a 100x lens (aperture 1.45), each sample underwent scanning. A blue monochromatic light ($\lambda = 460 \pm 5$ nm) was utilized for illumination. Imaging was conducted using a digital camera, SpotFex (Diagnostic Instruments), capturing images of twenty fields of view (FOV), each measuring 0.0156 mm$^2$.

For every FOV, 14 images were acquired, spanning 7 focus levels with 700 nm increments. Two polarizing configurations were applied: 1- Right circular polarization (RCP); 2- Left circular polarization (LCP), enabling the implementation of the Bidirectional Circular Polarization (BCP) method (Beaufort et al., 2021). The combination of RCP and LCP at each focus level was determined by the Eq. (1):

$$d = \frac{\lambda}{\pi \Delta n} \arctan\left(\sqrt{\frac{I_{LR}}{I_{LL}}}\right), \tag{1}$$




where d is the thickness, λ is the wavelength (460 nm), Δn is the birefringence of calcite (0.172), and $I_{LR}$ and $I_{LL}$ represent grey values measured with right and left circular polarizers, respectively.

After scaling the pixel values of the resulting images by multiplying them by 256 and dividing by 1.34, representing the maximum thickness measured at this wavelength, an 8-bit image was generated, ensuring compatibility with any image analysis program. To enhance focus for each small image segment, the seven images from each FOV were integrated using Helicon Focus software. This calibration process ensured that the light intensity in the resulting FOV image was fully adjusted for 3D imaging, guaranteeing sharp focus across all areas.

The precision achieved with this setup is remarkable, with measurements of 0.005 µm for thickness, and 0.032 pg/µm² for mass (Beaufort et al., 2021). Subsequently, the images underwent segmentation and analysis using an AI package called SYRACO, which combines morphometry and neural-network-based pattern recognition (Beaufort et al., 2022). SYRACO demonstrated its capability by providing precise mass and length measurements for *Emiliania huxleyi* coccoliths and coccospheres identified in the scans of the samples.


## 2.3. Coccolith Mass Estimations

Three methods were used to estimate individual coccolith mass: (1) based on EM analysis, (2) based on Coulter Multisizer ™ 4 coccolith volume analysis and (3) based on BCP. Coccolith mass from EM analysis was estimated according to the equation: mass = 2.7 * $k_s$ * $l^3$, where 2.7 is the density of calcite in g/cm³, $k_s$ is a constant dependent on the shape of the
coccolith and l is the distal shield length (Young and Ziveri, 2000). Shape constants were applied after Poulton et al. (2011) with $k_s$ = 0.02 and 0.015 for ecotype A and BC, respectively. An average number of 100 and 80 coccoliths were analysed for ecotype A and B/C, respectively. Average coccolith volumes obtained with the CM method were extracted from Müller et al. (2017) and transformed into the mass of coccolith by multiplying with the density of calcite. On average, approx. $6\times10^5$ coccoliths were measured per CM sample. Coccolith mass estimates derived from polarizing microscopy (bidirectional
circular polarization) analyses followed protocols described in Beaufort et al. (2014, 2021). An average number of 671 coccoliths were analysed per sample to estimate coccolith mass (see supplementary data).

## 2.4. Data and Statistical Analysis

Possible differences between the two studied strains of each ecotype were analysed by comparing the geometric parameters
and their linear regressions in regard to seawater $pH_t$ between the two individual strains of each ecotype using ANCOVA (p < 0.05). No significant differences were detected (Supplementary Table 1) and, consequently, the results of the individual strains of each ecotype were pooled and analysed as one data set for each ecotype. Variations of coccolith geometric parameters and mass in regard to seawater pH were analysed be means of linear regression analysis and significant



differences amongst the regression slopes were tested with the "statcalc" software (Soper, 2023). A significance level of 5 %

was applied for all statistical analyses.




# 3 Results

All coccolith geometric parameters of both ecotypes were significantly correlated with seawater $pH_t$ (Fig. 2 and
Supplementary Table 2). Increased coccolith size parameters (DSL, DSW, DSA, CAL, CAW and CAA) were associated to
lower seawater $pH_t$ values. The data sets of ecotype A and BC were significantly different from each other for all measured
geometric parameters (ANCOVA: $p < 0.05$). On the other hand, linear regression slopes of the relations between geometric
parameters and $pH_t$ were not significantly different between ecotype A and BC with the exception of CAL ($F = 5.60$, $p = 0.029$; Fig. 2e).

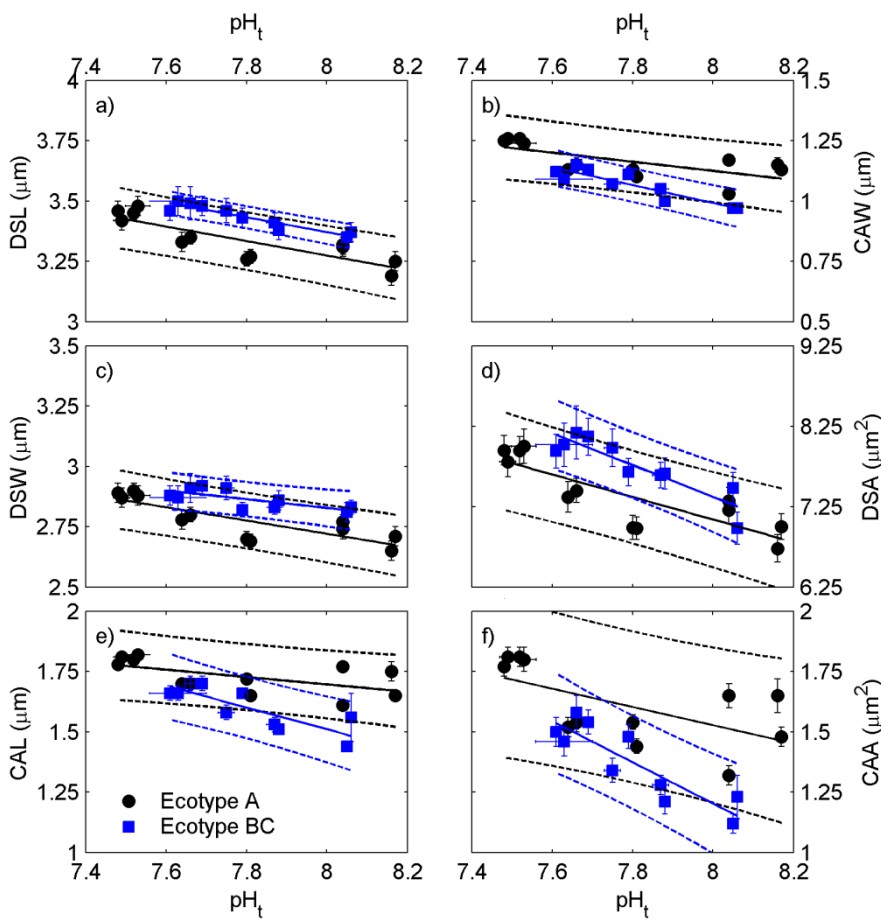


**Figure 2: Correlations of seawater carbonate chemistry, represented by pH(total scale), to coccolith (a) Distal Shield Length, (b) Central Area Width, (c) Distal Shield Width, (d) Distal Shield Area, (e) Central Area Length and (f) Central Area Area with corresponding standard error (se). Solid lines indicate significant linear regressions ($p < 0.05$) with 95% prediction intervals (dashed lines).**






Ecotype A coccolith mass estimations derived from EM, CM and BCP ranged from 1.83 to 2.37, 2.34 to 2.91 and 1.87 to 2.18 pg, respectively, and from 1.59 to 1.85, 2.08 to 2.36 and 1.31 to 2.25 pg, respectively, for ecotype BC. Changes in coccolith mass were significantly correlated to seawater $pH_{(total)}$ with the exception of coccolith mass estimations of ecotype BC derived from CM and BCP measurements (Fig. 3; Supplementary Table 3). The slopes of the linear regression lines of

the CM and BCP methods were not significantly different from the slope of the EM method ($p > 0.05$).

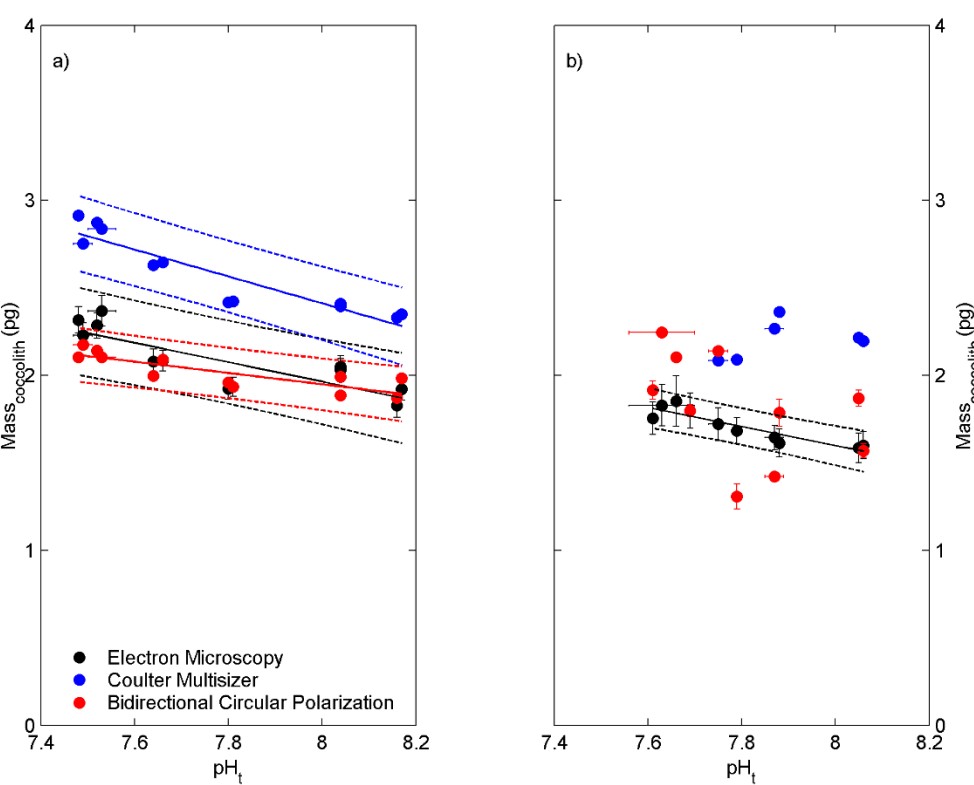

**Figure 3: Correlations of seawater carbonate chemistry, represented by $pH_t$, to average coccolith mass (±se) of ecotype A (a) and BC (b). Solid lines indicate significant linear regressions with 95% prediction intervals (dashed lines).**


Average coccolith masses were significantly different between the two ecotypes independent from the applied method (Table 1). Average coccolith mass from the CM method was significantly higher than the average values from the other two methods while no significant differences between the average values of the EM and BCP methods were detected (Table 1).






**Table 1: Average coccolith mass (± 1sd) of ecotype A and BC over the applied pH gradient. Significant difference between average coccolith mass values of each ecotype and differences amongst the applied method were detected by means of one-way ANOVA.**

| | Coccolith mass (pg) | | ANOVA[1] |
|---|---|---|---|
| | Ecotype A | Ecotype BC | p / F |
| EM | 2.09±0.18 (n=12) | 1.71±0.10 (n=10) | < 0.001 / 36.7 |
| CM | 2.58±0.22 (n=12) | 2.20±0.11 (n=6) | 0.001 / 15.6 |
| BCP | 2.02±0.10 (n=12) | 1.82±0.31 (n=10) | 0.042 / 4.7 |
| ANOVA[2] p / F | < 0.001 / 37.84[*] | < 0.001 / 10.96[*] | |

EM = Electron microscopy, CM = Coulter Multisizer, BCP = Bidirectional Circular Polarization. [1]ANOVA testing significant difference between ecotypes. [2]ANOVA testing significant differences amongst applied methods. [*]Mean coccolith mass value from CM analysis was significantly different from EM and BCP analyses while the difference between EM and BCP was non-significant (post-hoc Tukey HSD test).






## 4 Discussion

The results derived from EM indicate a clear influence of seawater carbonate chemistry on coccolith geometry and mass estimates (Fig. 2 and 3). The observation of higher coccolith mass in response to lower seawater pH may appear contradictory at first glance as frequently malformed coccoliths are associated with ocean acidification scenarios (e.g.,
Riebesell et al., 2000). However, this can be rationalized by considering that reduced calcification rates, induced by ocean acidification or reduced seawater pH, are independent from coccolith size (e.g., Müller et al., 2017). The detailed influence of seawater carbonate chemistry on coccolithophore physiology, calcification rates and coccolith geometry has been previously described (Bach et al., 2015; Meyer and Riebesell, 2015; Hermoso and Minoletti, 2018). Instead, this study is focused on the application of three different methods for estimating coccolith mass and the implications for future analyses.
The presented data include results from two ecotypes of *E. huxleyi* (i.e., A and BC) from the Southern Ocean. Ecotype BC has been described to produce relatively delicate coccoliths and demonstrates a high physiological sensitivity when exposed to changing environmental conditions compared to the more tolerant ecotype A (Cubillos et al., 2007; Cook et al., 2011; 2013). This is hypothesized to be related to the relatively constant ecological niche of type BC (i.e., the open Southern Ocean) and the evolutionary developed interplay of the underlying genetic and physiological framework (Schluter, 2009;
Chevin et al., 2014, Stotz, 2017). The high sensitivity to low seawater pH of ecotype BC resulted in diminished coccolith production and insufficient sample material for CM analysis at pH < 7.75 (Fig. 3b and d).

Three methods (i.e., EM, CM and BCP) were applied to estimate the coccolith mass. All these methods have been previously utilized to estimate coccolith masses from field and laboratory studies and the individual application procedures are well established and widely reported (see references in Table 2). The average coccolith masses (estimates from the three methods)
ranged from 1.59 to 2.91 pg, which is in good agreement with previous published values from laboratory and field studies (Table 2). Estimates > 6 pg are generally associated to sample material from laboratory experiments, presumably reflecting artificial culture conditions, leading to the formation of unusual large coccoliths (e.g., Müller et al., 2012).

**Table 2: Average estimates of *E. huxleyi* coccolith mass from laboratory, seawater and sediment samples. The method category PL**
**(Polarized light) includes different approaches to estimate the coccolith mass using polarizing light microscopy (please refer to each reference for a detailed description). EM = electron microscopy, CM = Coulter Multisizer, BCP = Bidirectional Circular Polarization, SYRACO = Système de Reconnaissance Automatique de Coccolithes (Beaufort and Dollfus, 2004).**

| Ecotype and/or origin[#] | Sample material | Mass (pg CaCO$_3$) | Method | Reference |
|---|---|---|---|---|
| Drake Passage | Sediment | 1.6 to 2.4 | EM | Vollmar et al. (2022) |
| Ecotype A / Patagonian Shelf | Seawater | 1.2 to 2.6 | EM | Poulton et al. (2011) |
| Ecotype A / Southern Ocean | Laboratory | 1.8 to 2.4 | EM | This study |
| Ecotype BC / Patagonian Shelf | Seawater | 0.9 to 1.9 | EM | Poulton et al. (2011) |





| | | | | |
|---|---|---|---|---|
| Ecotype BC / Southern Ocean | Laboratory | 1.6 to 1.9 | EM | This study |
| Ecotype A / Norway | Laboratory | 1.4 to 7.8 | CM | Müller et al. (2012) |
| Ecotype A / Southern Ocean | Laboratory | 2.3 to 2.9 | CM | This study |
| Ecotype BC / Southern Ocean | Laboratory | 2.1 to 2.4 | CM | This study |
| Canary Island | Seawater | 1.7 to 2.9 | PL | Linge Johnsen and Bollmann (2020) |
| Subantarctic Zone | Sediment trap | 1.8 to 4.3 | PL | Rigual-Hernández et al. (2020a) |
| South of Tasmania | Sediment | 2.2 to 3.3 | PL | Rigual-Hernández et al. (2020b) |
| Ecotype A / Southern Ocean | Laboratory | 1.9 to 2.2 | BCP | This study |
| Ecotype BC / Southern Ocean | Laboratory | 1.3 to 2.2 | BCP | This study |
| South Atlantic and Indian Oceans | Sediment | 1.7 to 4.9 | SYRACO | Horigome et al. (2014) |
| Mediterranean Sea | Seawater | 2.2 to 5.9 | SYRACO | D'Amario et al. (2018) |
| Mediterranean Sea | Sediment trap | 2.8 to 5.7 | SYRACO | Meier et al. (2014) |
| Ecotype A / Norway | Laboratory | 0.9 to 8.2 | SYRACO | Bach et al. (2012) |
| Ecotype BC / Atlantic Ocean | Laboratory | 1.5 to 5.1 | SYRACO | Beuvier et al. (2019) |

[#]Where no ecotype is specified, it is assumed that assemblages of *E. huxleyi* populations were reported containing a mixture of multiple ecotypes.

Coccoliths of ecotype BC have a more delicate structure than coccoliths of ecotype A and this is reflected in the lower coccolith mass estimates compared to ecotype A (Fig. 3a and b). This difference was detected to be significant independent from the applied method (Table 1), indicating comparable method sensitivities for different coccolith shapes of *E. huxleyi*. The non-significant differences between the linear regression slopes of ecotype A (Fig. 3a) further indicates the comparable sensitivity of the three methods to detects changes in coccolith mass induced by seawater carbonate chemistry variations.

However, the influence of seawater carbonate chemistry on the coccolith mass of ecotype BC was solely detected with the EM method which might be related to the coccolith sample size, orientation and morphological quality.

The ecotypes A and BC of *E. huxleyi* produce coccoliths that exhibit gaps lacking calcified structures. The size and number of these gaps are likely an interfering factor for the correct estimation of coccolith mass or $CaCO_3$ content. The shape constant ($k_s$, applied in the EM method) accounts for the general calcified structure of *E. huxleyi* coccoliths, and has been





estimated to vary between 0.016 and 0.021 for *E. huxleyi* ecotype A (Young and Ziveri, 2000). The shape constants for other *E. huxleyi* ecotypes are given with 0.015 (for ecotype BC) and 0.04 for the overcalcified form of ecotype A, where the central area (i.e., CAA) is completely calcified. Applying different shape constants to the linear regression models of Figure 3 generates a comparative overview of the influence of the coccolith shape and degree of calcification on the coccolith mass estimates from electron microscopy (Fig. 4). Interestingly, the coccolith mass estimates (ecotype A) from the CM method are

lower than estimates using the shape constant for the overcalcified coccolith structure ($k_s = 0.04$). This indicates that the CM method partly accounts for the structure of the coccoliths of ecotype A and BC and that possible occlusion (Plaats and Herps, 1984) of the analytical fluid (i.e., seawater) might be responsible for the observed mass overestimation in comparison to the EM and BCP methods. Here, a correction factor ($c_f$) of 0.8 is proposed to correct CM coccolith mass estimates to fit the estimates derived from EM and BCP (Fig. 4).


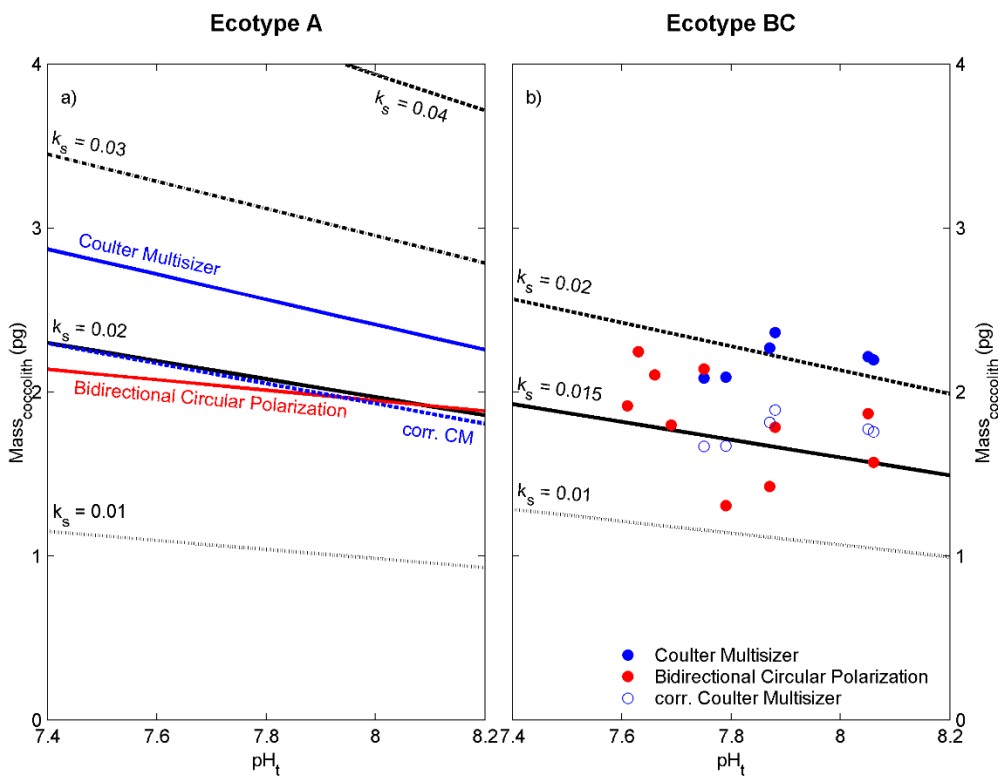

**Figure 4: Linear regression models or data points of coccolith mass estimates derived from Coulter Multisizer (blue) and cross-polarized light (red) compared to regression models from electron microscopy (black) with different coccolith shape constants ($k_s$) for ecotype A (a) and BC (b). Dashed blue line and open blue dots represent the corrected Coulter Multisizer estimates using the $c_f$**

**of 0.8 (see text for detail).**





Cross polarized light microscopy can provide information about the crystal orientation and structure of particles (Caamaño et al., 2010). It is relatively simple to use and does not require extensive sample preparation but may have limitations in terms of resolution and sensitivity compared to electron microscopy. The resolution of cross polarized light is constrained by the wavelength of applied light source (typically within the visible range). Consequently, accurately determining the size and shape of particles, particularly smaller ones or those with intricate morphologies (e.g., coccoliths), can be challenging and requires appropriate calibration due to the influence of sample thickness, orientation, and particle quality (Çopuroğlu, 2016). The calibration issues of polarizing microscopy have been overcome by the recently developed BCP method (Beaufort et al. 2021) and underlines the valuable application of polarizing microscopy for coccolith analysis, especially in the field of paleoceanography, where it allows for the examination of crystal orientation and structure of coccolith extracted from sediments. It is noteworthy that one should be cognisant of each method's limitations and consider employing complementary techniques for a more comprehensive and comparative coccolith analysis to introduce a possible correction factor for the diverse array of coccolithophore morphology and possible refractive properties. Instrumental costs and specific expertise required can differ substantially amongst the applied methods with BCP and EM relying on expensive equipment set-ups and on intense training and expertise compared to the CM method. In laboratory culture studies, however, the CM methods offers a fast and cost-effective estimation of coccolith mass from *E. huxleyi*.

In summary, the here presented results demonstrate that the methods utilized in this study are comparable in terms of their ability to detect alterations in coccolith mass caused by variations in ecotype specific structure and seawater carbonate chemistry. It should be noted, however, that the Coulter Multisizer estimates were slightly overestimated. In order to account for this discrepancy, a unitless correction factor ($c_f = 0.8$) has been proposed. The correction factor is applicable for both tested ecotypes of *E. huxleyi*. This adjustment allows for a high level of comparability between coccolith mass estimates obtained from the three methods and facilitates a future comparison and consolidation of mass changes observed from sediment, oceanographic and laboratory samples.



## 5 Conclusions

Three methods (Electron Microscopy, Coulter Multisizer and Bidirectional Circular Polarization) were applied to estimate the coccolith mass of two laboratory cultured populations of the coccolithophore *Emiliania huxleyi*. Average coccolith mass estimates are in good agreement with previous studies. However, coccolith mass estimates from Coulter Multisizer were
slightly overestimated and a correction factor has been introduced to compensate for this discrepancy. The relative change in coccolith mass triggered by ecotype specific structures and seawater carbonate chemistry are suitably captured by each of the three techniques and are comparable. In estimating the absolute values for coccolith mass, it is imperative to exercise prudence when employing the coccolith shape constants and refractory indices, with due consideration for the specific coccolithophore species and their corresponding morphological attributes.


**Declaration of Competing Interest:** The authors declare that they have no known competing financial interests or personal relationships that could have appeared to influence the work reported in this paper.

**Code/Data availability:** Research data is provided in the paper and the supplement.

**Author contribution:** CRV and MNM designed and conducted the study. LB executed the Bidirectional Circular Polarization analysis. CRV and MNM interpreted the data and wrote the paper with significant comments provided by LB and GMH.

**Acknowledgements:** We thank N. Hayes and J. Stone for capturing SEM images at UTAS. Funding to conduct and this study was provided by the Australian Research Council (ARC-DP grant 1093801) and the National Council for Scientific
and Technological Development (CNPq grant 305467/2020–4).



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
