# Peer review of "Technical note: A comparison of methods for estimating coccolith mass"

_EGUsphere, 2023_

## Author Response (AR1)

**Reviewer 1**

*Dear Reviewer 1,*

*Thank you for your review work and we appreciate your comments that certainly will improve our manuscript. Please find below your comments in normal font and my corresponding response in red font.*

The manuscript submitted by Valença and co-authors presents a comparison of different methods to estimate the mass of coccoliths, the calcite platelets produced by the coccolithophores, an important group of marine phytoplankton. The role played by these organisms in the global carbon cycle is well recognized, hence the need to accurately estimate the mass of calcium carbonate they produce. The three methods tested are the Scanning Electron Microscopy (EM), the Coulter Multisizer (CM) and the Bidirectional Circular Polarization (BCP). All three are established methods, the BCP being the most recent. It is worth noting that this is the first time such intercomparison is made. The material used (i.e.: the coccoliths) comes from a previously published laboratory experiment (Müller et al., 2015) allowing intercomparison between the different methods. In general, the manuscript is well written, with a valid scientific approach supported by good figures. The cited references are up to date and support the main results and conclusions. Although the manuscript is presented as a "Technical note" (see the title), its implications to better constrain the marine carbon cycle at different time scales, from paleo to present, makes it suitable for Biogeosciences. I would recommend this manuscript for publication in Biogeosciences after the authors addressed the minor revisions detailed in the following.

I appreciate the positive feedback.

**1 Introduction:**

p2, l.41: replace "Coccolith have been categorized" by "Coccoliths are categorized".

Corrected.

p3, l.54-80: in these paragraphs, the 3 methods are briefly described. Although the advantages of each method is presented, only the EM method appears to have some disadvantages/limitation (expensive, expertise needed, etc.). To be fair, the same should be mentioned for the 2 other methods.

**We added additional information to the Coulter (a) and the polarizing light methods (b), stating certain limitations:**

(a) "Coulter Multisizer sample material, however, requires liquid suspension and high particle numbers for appropriate analysis, limiting a generalized application."

(b) "Certain limitations, however, have been reported for polarized light microscopy and are associated with a particle thickness above 2 μm and with a vertical optical axis orientation of coccolith CaCO3 crystals (Beaufort et al., 2021)."

**2.1 Scanning Electron Microscopy:**

p4, l.100: Specify the type of filter used.

Filter material is specified in l. 93 as "Samples for EM and BCP microscopy were pooled for each triplicate treatment and filtered onto polycarbonate filters (0.8 μm pore size) and then dried at 60°C pending analyses."

p4, l.104-105: All the geometric parameters were measured from the pictures (EM) except DSA and CCA. This might be quite time consuming but a comparison between measured (with ImageJ) and estimated (ellipse area equation) would be welcome here. It could be performed on 100 random coccoliths (20 per pH treatment) of each ecotype. For example, the shape of the central coccolith shown in Fig. 1c is quite different compared to a "standard elliptical form". This might be even reinforced at lower pH where malformation is quite common.

This is a good point. Real DSA and CCA might indeed differ from their here calculated values. However, I would like to refrain from this investigation as it is not relevant to the current manuscript and the principal message. However, this research question can be further extended in a separate study, investigating/comparing different morphotypes of E. huxleyi. Presumably this will be addressed by a future graduate research project. I very much appreciate this comment and the underlying idea.

p5, fig. 1c: the picture of E. huxleyi ecotype A was published by Müller et al. (2015; Fig. 3e). Maybe the reference is needed there.

Reference has been included.

**2.2 Polarizing Microscopy:**

p5, l.115: Specify the type of filter used.

Filter material is specified in l. 93 (now l. 97) as "Samples for EM and BCP microscopy were pooled for each triplicate treatment and filtered onto polycarbonate filters (0.8 μm pore size) and then dried at 60°C pending analyses."

**2.3 Coccolith Mass Estimation:**

p6, l.137-138: "Coulter Multisizer TM 4", is it a typo?

Corrected.

**2.4 Data and Statistical Analysis**

p6, l.149 and on: The variations of coccolith geometric parameters were only estimated through scanning electron microscopy while the variations of coccolith mass were estimated through the 3 different methods. This should be clearly mentioned here.

This has been added now.

The average mass of coccolith estimated for the EM and BCP methods rely on the estimation of 80-100 coccoliths, while for the CM method, about $6x10^5$ coccoliths were measured. For clarity, the authors should briefly discuss it and show that the sample size may not affect the results, especially since the CM method shows significant different results compared to the 2 others.

The quantity of coccoliths analysed is reflected in the range/size of the standard error of the coccolith masses derived from each method (e.g., Fig. 3) and an insufficient sample size would certainly influence the interpretation of the results. Unfortunately, I am not sure which specific issue would need clarification. I apologize if I have miss-understood the raised point.

**3 Results:**

p8, l.161 and fig. 2: As mentioned before, the geometric parameters were estimated through the EM method. This should be mentioned here for clarity.

This has been included now.

p9, l.174: The statement about no significant differences between the slopes of the CM, the BCP and the EM methods only applies for the ecotype A here, or does it include as well the ecotype BC for the EM method? Please clarify.

Yes, the statement is only valid for ecotype A. This has been stated now.

p9, l.182: See the previous comment on the sample size.

As in the previous comment, I am indecisive on how to address this issue. However, it should be mentioned that in this section the total average over the applied pH gradient was compared amongst the ecotypes and methods which has been stated now in the text. I hope this might help to clarify this point.

**4 Discussion:**

p11, l.206: Fig. 3b (there is no Fig. 3d).

Corrected.

p11, l.210: Since in table 2 a distinction is made between the 2 ecotypes (when possible), it would be appropriate to present the average ranges for them in the main text.

Good point. The coccolith mass ranges for each ecotype have been included now to the main text.

p13, l.238: A brief description of how the correction factor was estimated would be appreciated here.

The correction factor was estimated by fitting the linear regression (ecotype A) of the CM estimates to intersect the linear regression of the EM estimates at a pH of 7.8. This resulted in a cf value of 0.808. However, the value of 0.8 has been applied for simplicity and clarity. This is now stated in the according section of the manuscript as: "A correction factor of 0.808 was calculated to fit the estimates derived from EM and BCP by fitting the linear regression of the CM estimates (Fig. 3a) to intersect the linear regression of the EM estimates at pHt of 7.8. Here, a simplified/rounded correction factor (cf) of 0.8 is proposed to be applied to CM coccolith mass estimates (Fig. 4).

*Dear Prof. Poulton,*

*Your review is much appreciated and improved our manuscript with additional information and expertise. Please find below your comments in normal font and my corresponding response in red font.*

**General:**

An interesting and important paper comparing the common methods for determining coccolith mass for coccolithophores in the ocean, laboratory and sediments. Consolidates and clearly recognizes the benefits of the three different methods and makes an unbiased assessment of their relative merits. One issue that needs clarifying in the text, for those unaccustomed with such measurements, is that the reported values of coccolith mass are CaCO3 mass (pg CaCO3); often carbon mass (pg C) is of interest and so the authors should make this clear – it was only in Table 2 that this was made obvious to the reader.

Thanks for the positive evaluation of our study. The mass unit of coccoliths is a very important and essential point and I made it now clearer throughout the manuscript that coccolith mass values are expressed in pg CaCO3.

**Specific Comments:**

Ln 12: Coccolithophore parameters are relevant to more than just paleo-proxy studies, and this is only one example of their interest and importance to ocean sciences. Please rephrase this introductory line to better reflect this.

Additional information has been added to the abstract: "Variations in coccolithophore parameters (e.g., community composition, morphology, size and coccolith mass) are a key factor for ocean biogeochemical dynamics (e.g., biological carbon pump) and have been used as paleoproxy to understand past oceanographic conditions."

Ln 29: Sviben et al. (2016) – using this as a reference to support the point that coccoliths contribute to the global carbon cycle seems odd. The reference is a physiological and cellular ultrastructure study of coccolith formation rather than a paper establishing coccolithophore contributions to the carbon cycle. See (e.g.) Neukermans et al. (2023) Quantitative and mechanistic understanding of the open ocean carbonate pump – perspectives for remote sensing and autonomous in situ observation. Earth-Science Reviews 239, doi: 10.1016/j.earscirev.2023.104359.

Agreed and reference has been replaced.

Ln 32: Jin et al. (2019) – again, this is a slightly odd choice of reference as the study by Jin et al. (2019) is an examination of sediment trap material and not an 'accurate' comparison of upper ocean communities with those found in underlying sediments (the point made in the current

paper). This ignores important potential taphonomic changes. Further, this point does not really add to the paper by Valenca et al., so is it needed?

No, it is not needed. Statement was removed from the manuscript.

Ln 36: <coccolithophores> rather than <coccolithophore>.

Corrrected.

Ln 40: Sucheras-Marx et al. (2022) does not appear in the reference list. Correct citation?

Reference has been added.

Ln 55: See also Charalampopoulou et al. (2016) (though see comment on Table 2).

Reference has been added and data included to Table 2.

Ln 56: <coccoliths impedes a direct measurement of their mass> rather than <coccoliths impedes a direct measurement of its mass>.

Corrected.

Ln 62: Saruwatari et al. (2011) – there are many examples of studies to support the point that EM reveals coccolith morphology so suggest that this is cited as (e.g., Saruwatari et al., 2011).

Corrected.

Ln 72-73: This sounds rather cryptic – does having sufficient material to determine extracellular CaCO3 content refer to elemental analysis (CHNS) and determination of particulate carbon? Please clarify this line.

This phrase was reformulated for better clarity: "The Coulter Multisizer principle provides a rapid method to count coccolithophore cells and coccoliths (e.g., Müller et al., 2021), and to estimate extracellular CaCO3 content (Fan et al., 2022) from laboratory samples."

Ln 13: Odd word choice - 'remarkable' – please use a more scientific term or rephrase.

The phrase was adjusted to: "This calibration process ensured that the light intensity in the resulting FOV image was fully adjusted for 3D imaging, guaranteeing sharp focus across all areas, achieving an accuracy of 0.005 µm for thickness and 0.032 pg/µm² for mass (Beaufort et al., 2021)."

LN 133: When using the word 'precise', how can you be sure that these are 'exact and accurate' when combining three different techniques? Precise infers that these are highly accurate (i.e. close to the true value), might it be better to term them 'accurate' (i.e. they are in self agreement with one another).

Agreed and corrected.

Ln 160: Worth citing Figure 1 to remind the reader of the definitions of these different terms?

Good point. Included.

Table 1: Coccolith mass (pg CaCO3) rather than Coccolith mass (pg); as is done in Table 2.

Corrected and adjusted also for axis labels of Fig. 3 and 4.

Table 2: Charalampopoulou et al. (2016) also give similar values from EM of ecotype BC across Drake Passage of 0.6 – 1.5 pg CaCO3; which agree with other values given in this table and provides another Southern Ocean study to compare with. Also, might be worth pointing out that Drake Passage is in the Southern Ocean. (Charalampopoulou et al. 2016 Environmental drivers of coccolithophore abundance and calcification across Drake Passage (Southern Ocean). Biogeosciences, doi: 10.5194/bg-13-5917-2016).

I have now included Charalampopoulou et al. (2016) to Table 2 and appreciate the reference as data on ecotype BC are rather rare.

Ln 222: Please explain 'significant independent from the applied method'. Is this statistically independent (p<0.001) from the applied method, or is a word missing?

I rearranged the phrase for better clarity to: "This difference was detected to be significant (p < 0.05), independent from the applied method (Table 1), indicating comparable method sensitivities for different coccolith shapes of *E. huxleyi*."

Ln 231-232: Please give a reference for the shape constants for BC and A coccolith shape constants.

References included.